# Photoprotective effect of *Astragalus membranaceus* polysaccharide on UVA-induced damage in HaCaT cells

**Qiong Li**[1,2], **Depeng Wang**[1,2], **Donghui Bai**[1,2], **Chao Cai**[1,2,3], **Jia Li**[1,2], **Chengxiu Yan**[1,2], **Shuai Zhang**[1,2], **Zhijun Wu**[4], **Jiejie Hao**[1,2,3]*, **Guangli Yu**[1,2,3]*

1 Key Laboratory of Marine Drugs, Ministry of Education, School of Medicine and Pharmacy, Ocean University of China, Qingdao, China, 2 Shandong Provincial Key Laboratory of Glycoscience and Glycotechnology, School of Medicine and Pharmacy, Ocean University of China, Qingdao, China, 3 Laboratory for Marine Drugs and Bioproducts, Pilot National Laboratory for Marine Science and Technology (Qingdao), Qingdao, China, 4 Infinitus (China) Company Ltd., Guangzhou, China

* 2009haojie@ouc.edu.cn (JH); glyu@ouc.edu.cn (GY)

**Data Availability Statement:** All relevant data are within the manuscript and its Supporting Information files.

## Abstract

### Background

The skin provides a predominant barrier against chemical, physical and microbial incursion. The intemperate exposure to ultraviolet A (UVA) radiation can cause excessive cellular oxidative stress, leading to skin damage, proteins damage and mitochondrial dysfunction. There is sufficient evidences supporting the proposal that mitochondria is highly implicated in skin photo-damage.

### Methods

In the present study, a polysaccharide isolated from *Astragalus membranaceus* was further purified to be an α-glucan, which was further investigated its beneficial influence on UVA-induced photo-damage in HaCaT cells.

### Results

Our results showed that the purified *Astragalus membranaceus* polysaccharide (AP) can protect HaCaT cells from UVA-induced photo-damage through reducing UVA-induced intracellular ROS production and mitochondrial membrane potential, thereby altering ATP content. It was found that the UVA induced damage in HaCaT cells could be effectively restored by co-treatment with AP.

### Conclusions

AP exhibited promising potential for advanced application as multifunctional skin care products and drugs.

**Funding:** This work was supported by National Natural Science Foundation of China (81973231, 81402982), National Science and Technology Major Project for Significant New Drugs Development (2018ZX09735004), Taishan Scholar Project Special Fund (TS201511011). The funders had no role in study design, data collection and analysis, decision to publish, or preparation of the manuscript.

**Competing interests:** The authors of Qiong Li, Depeng Wang, Donghui Bai, Chao Cai, Jia Li, Chengxiu Yan, Shuai Zhang, Jiejie Hao and Guangli Yu declare that they have no competing interests; Zhiyun Wu also provides a Competing Interests Statement declaring that she has no competing interests.

**Abbreviations:** MMP, mitochondrial membrane potential; JC-1, J-Aggregate forming lipophilic cation 55′,6,6′-tetrachloro-1,1′,3,3′ tetraethylbenzimidazolcarbocyanine iodide; BSA, bovine serum albumin; ddH2O, double-distilled water; AP, Astragalus membranaceus polysaccharide; DCFH2-DA, 2',7'-dichlorofluorescin diacetate.

## Introduction

As well all know that UV radiation is the main factor that accelerates skin photo-damage and accelerates the natural aging process. It has been fully reported that 90% of skin damage is caused by UV radiation [1–3]. The UV scope consists of three wavelength ranges: UVC (200–280 nm), UVB (280–320 nm) and UVA (320–400 nm) [4]. 90% of the solar radiation reaching the Earth's surface is composed of UVA, and it penetrates the skin deeper than UVB [5, 6]. And long-term radiation to UVA radiation can cause numerous skin lesions, such as photo-sensitivity skin diseases and cancer [7].

The connection of aging phenotypes to cellular senescence and dysfunction has been reported in many researches [8–10]. Mitochondria can produce ATP which is used as chemical energy for most eukaryotic cells, and they also control cell functions related to differentiation, cell signaling, cell growth and death [11].Mitochondria are the main source of ROS production though OXPHOS activity in the inner mitochondrial membrane (mainly at complex I and II), which in turn is the major target of ROS damage [11, 12].

Mitochondrial DNA (mtDNA) attaches to the matrix of the inner membrane of the mitochondria, which is extremely sensitive to ROS in the mitochondria. Meanwhile, it was also reported that our body's reactive oxygen species (ROS), mitochondrial dysfunction and mitochondrial DNA (mtDNA) damage increased with photo-damage [13, 14]. UVA radiation could induce mitochondrial dysfunction which directly contributes to photo-damage, and lead to the generation of mutations in mitochondrial [15, 16]. ROS still act on cardiolipin (CL), which can be oxidized and then converted into lipid second-messengers to advocate oxidative stress and disrupt mitochondrial metabolism [17, 18]. Mitochondrial dysfunction can induce cellular calcium influx and decreases ATP production, which could influence the cell membrane permeability in a feedback loop and mitochondrial membrane potential [19]. Indeed, there is increasing interest in protecting mitochondria from UVA induced ROS damage for the precaution and treatment of UVA-induced photo-damage. Consequently, a number of recent researches discovered that the plant extracts and natural products are particularly effective in the protection of skin from photo-damage by targeting to mitochondria, such as proanthocyanidins, carotenoids, lycopene [20–23]. Some reports have reported that glucan how potent inhibitory effects against UV-induced skin photo-ageing, which was induced by UVB irradiation through ROS. Such as alpha1-4-glucan, β-1,3/1,6-glucan and β-(1,3)-β-(1,4)-glucan [24–26].

*Astragalus membranaceus* is a traditional Chinese herbal medicine. Many studies have shown that *Astragalus membranaceus* exhibits multiple physiological functions that confer anti-oxidation, anti-inflammatory, anti-diabetic and anti-tumor. Among them, Astragalus flavonoids, saponin and polysaccharides are investigated to be the main active ingredients that exert those diverse functions [27, 28]. Furthermore, polysaccharides, as one of the major components in *Astragalus*, have also been widely investigated for their significant bioactivities. However, there is no systematic research previously reported the anti-photo-damage effect of *Astragalus* polysaccharide. Skin health promotion effects of natural glucan derived from cereals and microorganisms have been studied before, but their functional mechanism have rarely reported especially through mitochondrial mediation.

In our study, a representative polysaccharide was isolated and purified from *Astragalus membranaceus*. The total sugar, protein content and molecular weight ($M_w$) of AP were characterized to be 64.9%, 20.1% and 32 kDa, respectively. The structural character of major component of AP was identified to be α-(1→4)-glucan by infrared, 1D and 2D-NMR spectra. The protective effects of AP on UVA-induced photo-damage HaCaT cells was evaluated.

## Materials and methods

### Materials and reagents

*Astragalus membranaceus* (from Gansu Province) was provided by Infinitus (China) Company Ltd. (Guangzhou, China). MEM medium, fetal bovine serum (FBS) were purchased from Hyclone (Logan, UT), penicillin Invitrogen (Carlsbad, CA) and streptomycin were purchased from Gibco (Grand Island, NY).

### Extraction and purification of AP

The dried roots of *Astragalus membranaceus* were pulverized and sieved through 60-mesh to obtain fine powder. Deionizer water was added to *Astragalus* powder at the ratio of 20/1 (v/m), extracted by reflux at 80˚C. The mixture was centrifuged at 0˚C, and then the supernatant was collected. Then, deionized water was added to the residue at the ratio of 15/1 (v/m). The mixture was extracted repeatedly by reflux at 80˚C for 3h and centrifuged again. The supernatant was combined and concentrated by using a vacuum rotary evaporator. Crude polysaccharide was obtained after precipitation with four volumes of absolute ethanol and freeze-dried.

The crude polysaccharide was purified on the Q-Sepharose Fast Flow (QFF) column with gradient elution of 0–2 M NaCl solutions at a flow rate of 3.0 ml/min. The purified polysaccharide was collected at 0 M NaCl, which was named as *Astragalus membranaceus* polysaccharide (AP).

### Total sugar and protein contents of AP

The total sugar content in AP was measured by using the standard phenol-sulfuric acid method [29]. The protein content in AP was determined by using the Coomassie Brilliant Blue method [30].

### Molecular weights and monosaccharide composition analysis of AP

The weight-averaged molecular weight ($M_w$) of AP were determined by high performance gel permeation chromatography combined with multi-angle laser light scattering (HPGPC-MALLS) on an Agilent 1260 chromatographic instrument. The AP was dissolved in 0.1 mol/L $Na_2SO_4$ solution at a concentration of 5 mg/ml. Next, the AP(100ul) was added to a Shodex OHpak column(35˚C) and washed with 0.1 mol/L Na2SO4 solution at 0.6 ml/min. And then, the signal was detected by G1362A refractive index detector (RID) and MALLS (Dawn Heleos-II, Wyatt technologies, USA). The $M_w$ of AP was calculated by Astra 5.3.4.20 software.

Monosaccharide composition of AP was determined by a pre-column derivatives 1-phenyl-3-methyl-5-pyrazolone(PMP)-high performance liquid chromatography (HPLC) method. The polysaccharide was hydrolyzed by TFA. Then, the sample was heated with PMP at 70˚C for 1h. The sample was analyzed by using a $C_{18}$ column at 1 ml/min, which was determined at 245 nm.

### Fourier Transform in Infrared Spectroscopy (FT-IR) and Nuclear Magnetic Resonance (NMR) analysis of AP

For FT-IR analysis,100 mg dried KBr was mixed with the dried AP (3–4 mg) and then monitored by a Nexus 470 FT-IR spectrometer (Thermo Electron) at 400–4000 cm−1 under dry air. For 1H-NMR and $^{13}$C-NMR analyses, the AP was dissolved in D2O, and t freeze-dried twice.

Spectra were analyzed by using JNM-ECP 600 MHz equipment (JNM-ECP 600, Jeol, Japan) at 25˚C. The data was analyzed by using the MestReNova software.

## Cell culture and treatments

HaCaT cells, from American Type Culture collection (ATCC), were grown in a humidified (5% CO2, 95% air) atmosphere with MEM medium containing 10% fetal bovine serum, and protected by 2 mM L-glutamine, 100 U/ml penicillin/streptomycin. HaCaT cells seeded in 96-well micro plate were cultured at 37˚C for 12h. Then, the HaCaT cells grown in MEM supplemented with various concentrations of AP at 37ºC and 5% $CO_2$ for 48 hours. The medium was discarded, serum-containing MEM was added and exposed in 30 J/cm$^2$ ultraviolet irradiation.

## MTT assay

After the cells were treated for 24h, the HaCaT cell viability was analyzed by using MTT assay. Then, 20μl/well of MTT solution (5 mg/ml in PBS buffer) was added for 4h. The medium was aspirated, and then replaced with DMSO for dissolving the Potassium salt. The color intensity of the Potassium solution, which reflects the cell growth condition, was measured at 570 nm using a micro plate spectrophotometer [31, 32].

## Assay for Mitochondrial Membrane Potential (MMP, ΔΨm)

Treated HaCaT cells with a dual emission potential-sensitive JC-1 probe (5,5',6,6'- tetrachloro-1,1',3,3'-tetraethyl-benzimidazolyl-carbocyanine iodide) in 96-well plates for 30 min at 37˚C [33]. And then, determined by a dual-wavelength/double-beam recording spectrophotometer (Ex490nm/Em530, Ex525nm/Em590, Flex Station384, Molecular Devices, USA).

## Detection of ROS

The content of cellular ROS was determined by 2',7'-dichlorofluorescin diacetate (DCFH2-DA) [34]. In short, HaCaT cells were incubated with 2μM DCFH2-DA for 45 min at 37˚C. Next, washed with PBS buffer four times. Finally, the fluorescence intensity was measured by using spectrophotometer (Flex Station 384, Molecular Devices, USA) 485 nm/535 nm [35, 36].

## Determination of ATP content

After various treatments, HaCaT cells were solubilized by Triton X-100(0.5%) with Glycine buffer (100 mM pH 7.4). Intracellular ATP levels were measured by an ATP bio-luminescence assay kit (Sigma) [37]. ATP levels are expressed as micromoles per cell.

## Activity of mitochondrial complexes I, II

HaCaT cells were grown in 100 mm plates. The HaCaT cells were collected by centrifugation at 1500 g for 10 min at 0˚C. The collecting pellet was resuspended in mitochondrial isolation buffer (215mM mannitol, 75mM sucrose, 0.1% BSA, 1 mM EGTA, 20mM HEPES, pH 7.2), and then homogenized on ice by a glass homogenizer. The supernatant fraction was centrifuged at 13000 g for 10 min to pellet the mitochondria. Briefly, complex I activity was measured by following the change in NADH for 10min at 340 nm. Next, complex II was determined by using mitochondria (final concentration:30 mg/ml), and then the reaction was started with 10 mM succinate and scanned at 600 nm for 2 min.

**Table 1. General structural characterization of AP.**

| Sample | Total sugar content | Protein content | $M_w$ | $M_w/M_n$ | Monosaccharide molar ratio (%) | | |
|---|---|---|---|---|---|---|---|
| | | | | | Glc | Man | Gal |
| AP | 64.9% | 20.1% | 32kD | 1.74 | 98.33 | 0.33 | 1.34 |

## Results

### Preparation and structural characterization of AP

The crude AP was prepared from the dried roots of *Astragalus* membranaceus by using hot water extraction. The crude AP was subsequently separated by Q-Sepharose Fast Flow (QFF) column at 0 M of NaCl to obtain the purified AP. The structural properties of AP including sugar and protein contents, molecular weights ($M_w$), and monosaccharide compositions were fully characterized. As depicted in Table 1, the total sugar and protein contents of AP were calculated to be 64.9% and 20.1% respectively. The $M_w$ of AP was determined to be 32 KD by HPGPC-MALLS, and the symmetric peak confirmed the purity of the polysaccharide. As shown in Fig 1, PMP-HPLC method was employed to obtain the monosaccharide compositions of AP, and its main component was identified to be glucose with small amounts of mannose and galactose in the ratio of 98.33: 0.33: 1.34 (Glc: Man: Gal).

The FT-IR spectrum of AP is shown in Fig 2(A), the bands at around 3367 cm$^{-1}$ were assigned to symmetric stretching vibration of the hydroxyl groups. The band at 2933 cm$^{-1}$ was attributed to C–H stretching vibration, while the band at around 1417 cm$^{-1}$ was attributed to C–H bending vibration. The polysaccharide hydration vibration absorption peak appeared at 1643 cm$^{-1}$. Meanwhile, the characteristic absorption bands of polysaccharide were found at

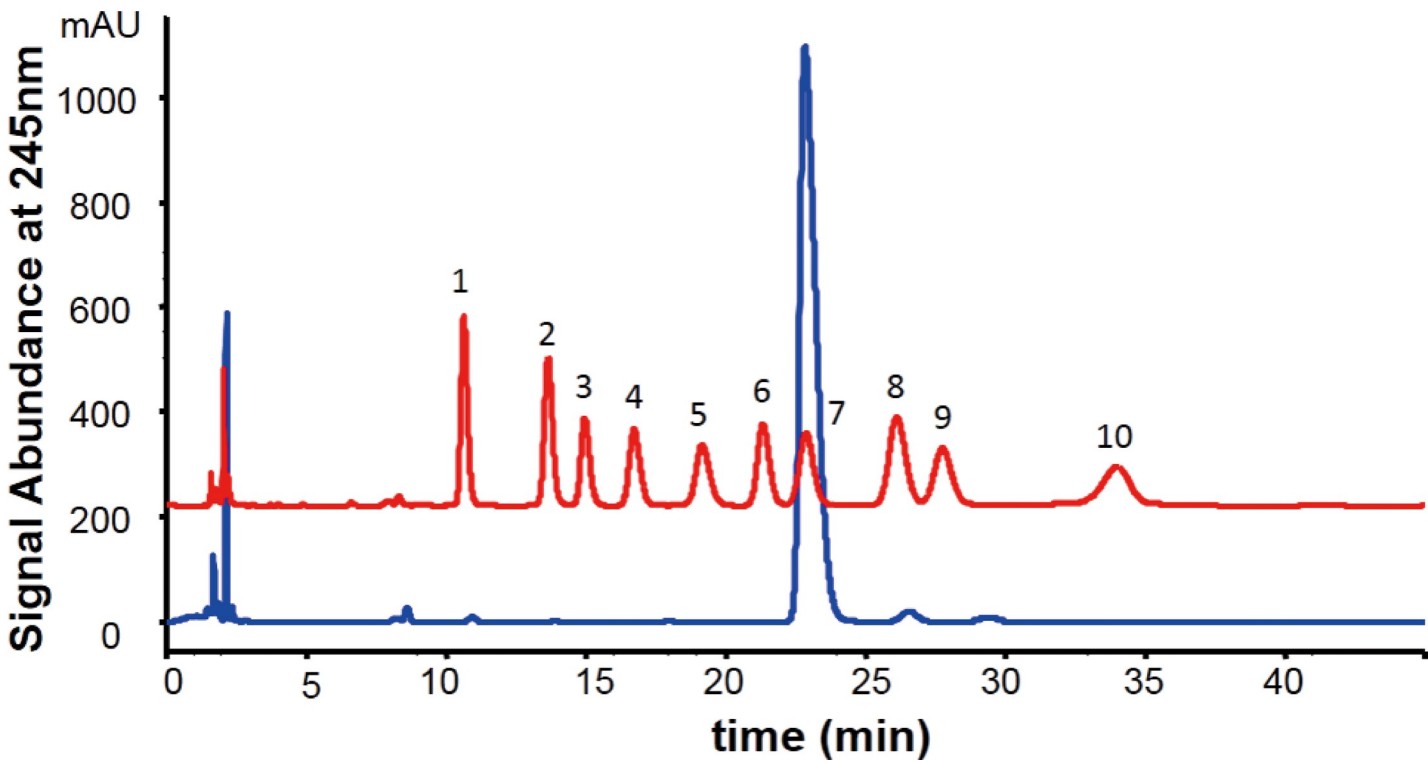

**Fig 1. The monosaccharide composition analysis of AP.** (1. Man, 2. GlcN, 3. Rha, 4. GlcA, 5. GalA, 6. GalN, 7. Glc, 8. Gal, 9. Xyl, 10. Fuc).

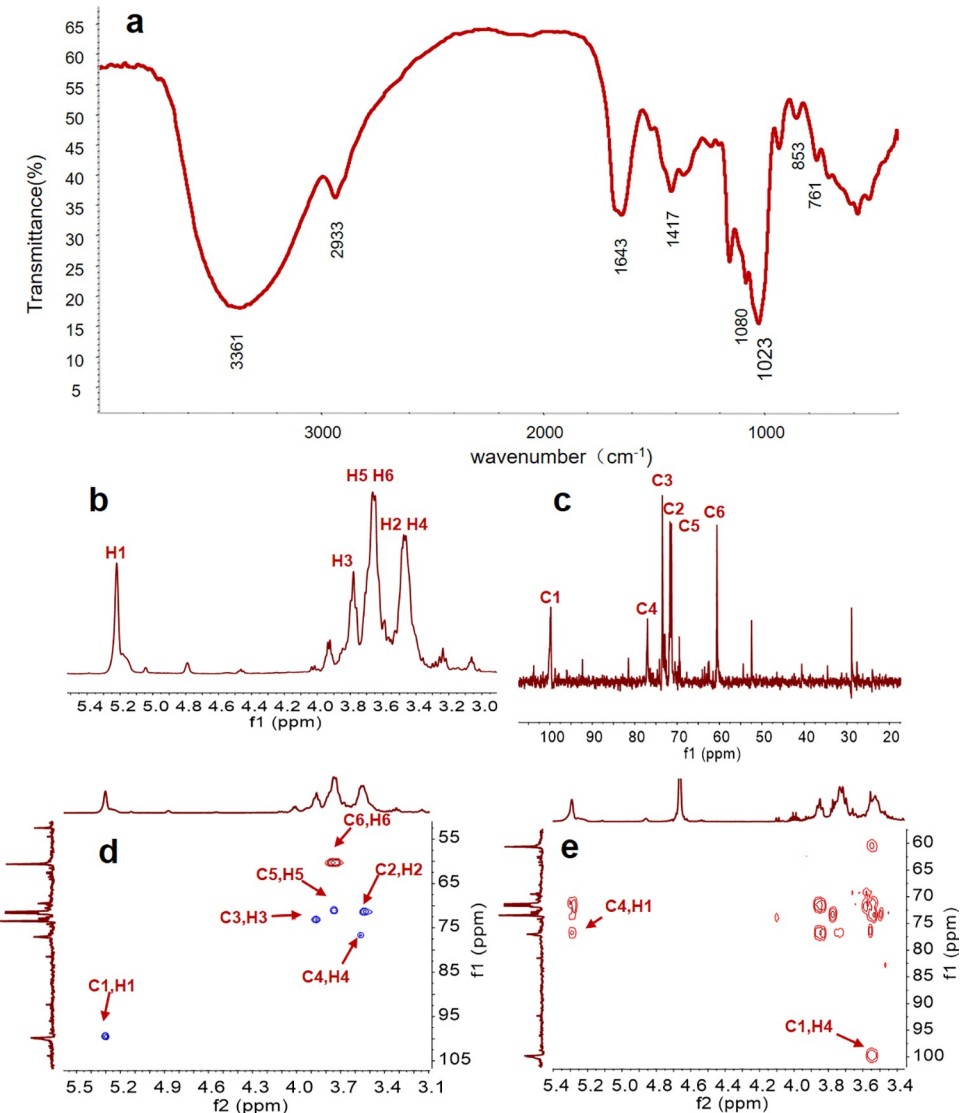

**Fig 2. The FT-IR and NMR analysis.** (a) FT-IR spectrum of AP. (b) ¹H-NMR spectrum of AP. (c) ¹³C-NMR spectrum of AP. (d) ¹H-¹³C HSQC spectrum of AP. (e) ¹H-¹³C HMBC spectrum of AP.

around 1080 cm⁻¹ for C–O–C stretching and 1023 cm⁻¹ for C–O stretching. The band at around 853 cm⁻¹ was the characteristic absorption of glycosidic bond. The band at around 761 cm⁻¹ was stretching vibration of glucose pyranose rings. The structure of AP was further characterized by NMR spectroscopy, and the chemical shifts of protons and carbons were well assigned accordingly. As shown in Fig 2(B), the apparent single peak at 5.22 ppm and minor double peak at 4.91–4.94 ppm corresponded to the H1α and H1β at the reducing end. It is obviously observed that the AP is mainly composed of α configuration. The peak at around 99.80 ppm corresponded to the C1 signals in the ¹³C NMR spectra (Fig 2(C)) of AP. The peaks at 71.69, 73.47, 77.01, 71.35 and 60.63 ppm were assigned to C-2, C-3, C-4, C-5 and C-6, according to ¹H-¹³C HSQC spectrum (Fig 2(D)). In addition, hydrocarbon remote correlations between C4 and H1 on α-glucan were also clearly identified in ¹H-¹³C HMBC spectrum

(Fig 2(E)). In summary, the AP was preliminarily identified to be an α-(1→4)-glucan after fully structural characterization.

## Cell toxicities of AP

As shown in Fig 3, the cell toxicities of AP on HaCaT cells was detected by MTT assay. It was demonstrated that AP has no toxic effects on HaCaT cells at concentrations ranging from 50–600μg/ml (Fig 3(A)). HaCaT cells were exposed to different doses of UVA as indicated. Meanwhile, as shown in Fig 3(B), pretreatment of HaCaT cells with 200μg/ml, 400μg/ml, 500μg/ml significantly increased the cell viability. Furthermore, there was no significant increase in the MTT assay of UVA-induced HaCaT cells at a concentration of 600μg/ml compared to 200μg/ml. Therefore, we chose 50–200μg/ml to continue our ΔΨm detection and changes in intracellular ROS content.

## Effect of AP on Mitochondrial Membrane Potential (MMP)

We performed a JC-1 assay to determine changes in mitochondrial membrane potential induced by different doses of UVA. As shown in Fig 4, pretreatment of HaCaT cells with different concentrations of AP significantly inhibited the decrease of ΔΨm in UVA-induced HaCaT cells as determined by JC-1 assay. Within the range of 50–600μg/ml, AP could increase the mitochondrial membrane potential, especially at the concentrations of 100–600μg/ml. In addition, at the concentration of 200μg/ml and 500μg/ml of AP are preferred.

## Effect of AP on ROS production and ATP content

Then ROS production was determined and the results are shown in Fig 5(A). UVA exposure of 30 J/cm$^2$ resulted in significant increase in ROS production in HaCaT cells, and pre-treatment with AP significantly inhibited ROS production.

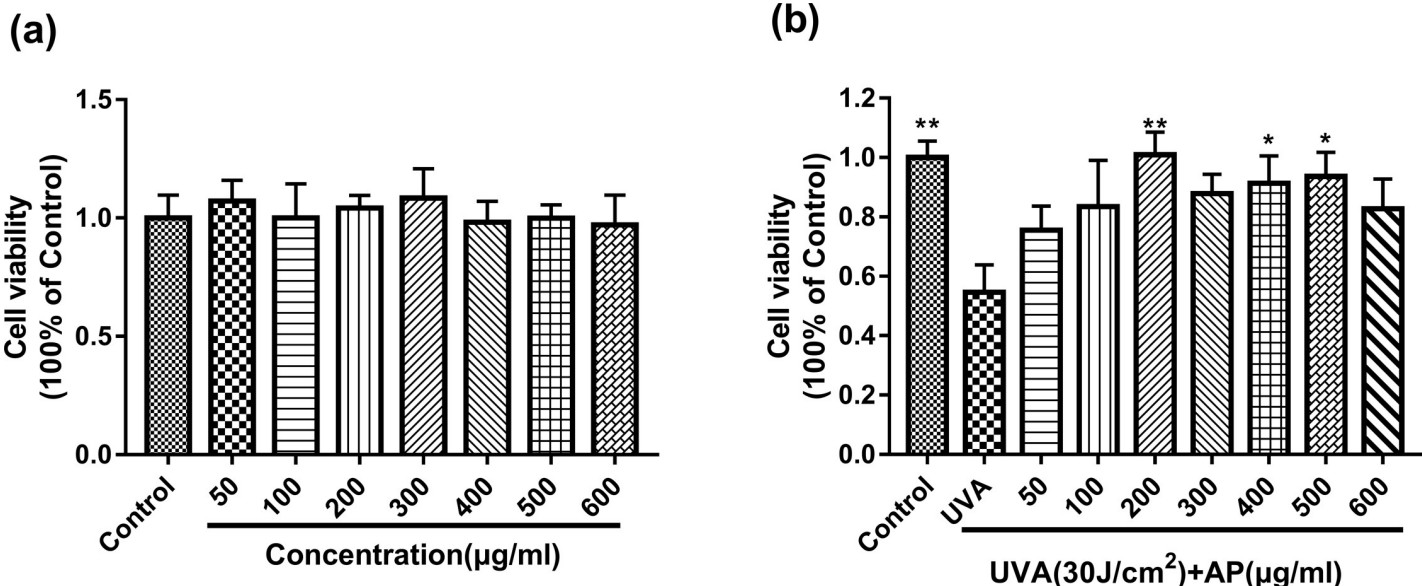

**Fig 3. Cytotoxicity of AP on HaCaT cells.** (a) Cells were incubated with different concentrations of AP for 48 hours. (b)The cells were pretreated with 0–600μg/ml AP for 48h, and then irradiated with UVA(30J/cm$^2$).Cell viability was assessed by MTT assay as described in Materials and Methods. Values are the mean of nine replicates ± S.E.M. $^*P$ <0.05 and $^{**}P$ < 0.01 mean significant difference compared to the UVA group.

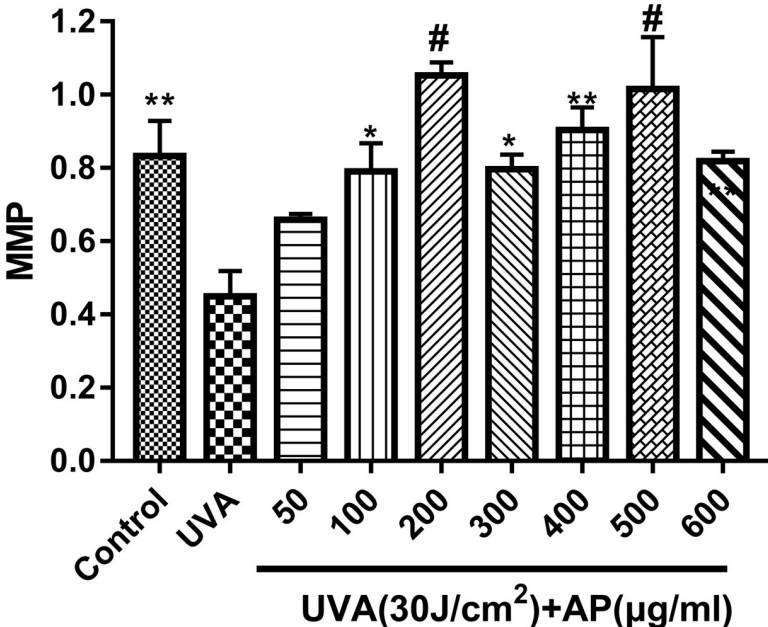

**Fig 4. Effect of AP on ΔΨm in UVA-induced HaCaT cells.** HaCaT cells were incubated with the indicated concentrations of AP for 48 hours and then irradiated with UVA (30J/cm²). After 12 hours, the cells were washed and incubated with JC-1 for ΔΨm assay. $^*P < 0.05$, $^{**}P < 0.01$ and $^{\#}P < 0.0001$ mean significant difference from UVA group. Error bars are expressed as mean ± S.E.M. ($n = 9$).

As shown in Figs 3 and 4, the optimal concentration of AP to protect HaCaT cells is 200μg/ml or 500μg/ml. Fig 5(A) shows that the addition of 200μg/ml significantly reduced ROS levels. Fig 5(B) shows that UVA (30J/cm²) significantly reduced ATP levels, and 200μg/ml AP

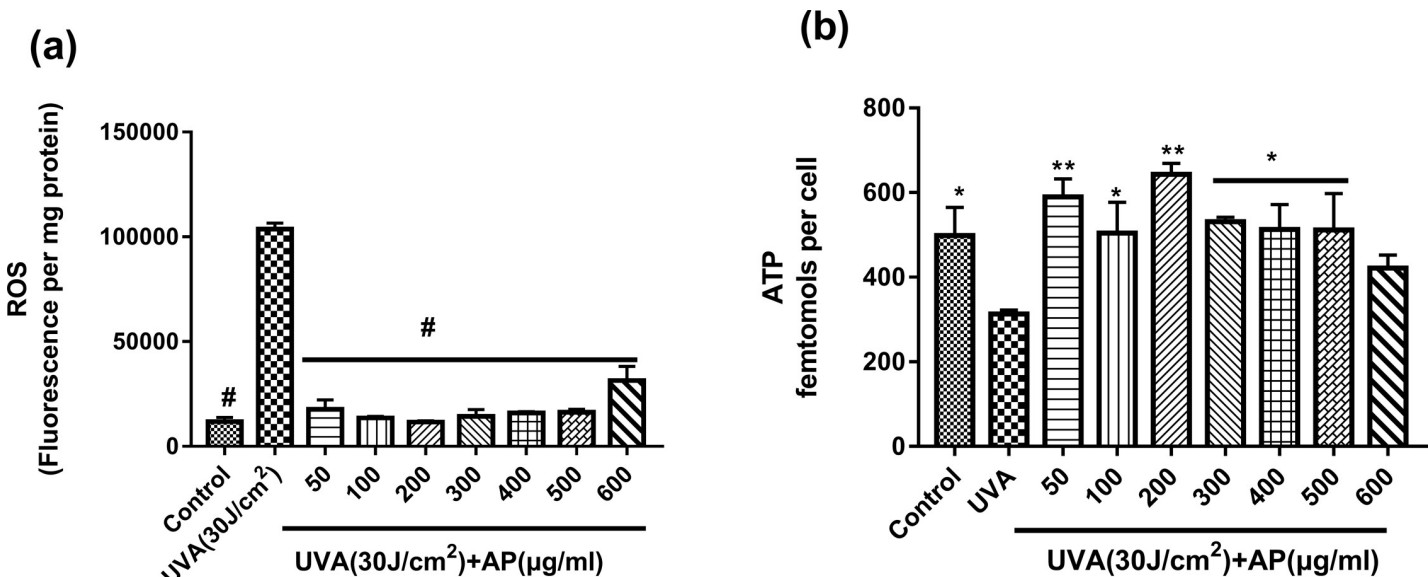

**Fig 5. Effect of AP on ROS production and ATP content in UVA-induced HaCaT cells.** Cells were incubated with the indicated concentrations of AP for 48 hours and then irradiated with UVA (30 J/cm²). After 12 hours, the cells were washed and incubated with CFH2-DA for ROS production assay (a); or with luciferase-based luminescence assay kit for ATP content measurement (b). Values are the mean ± S.E.M of data from at least nine independent experiments. $^*P < 0.05$, $^{**}P < 0.01$ and $^{\#}P < 0.0001$ compared to the UVA group.

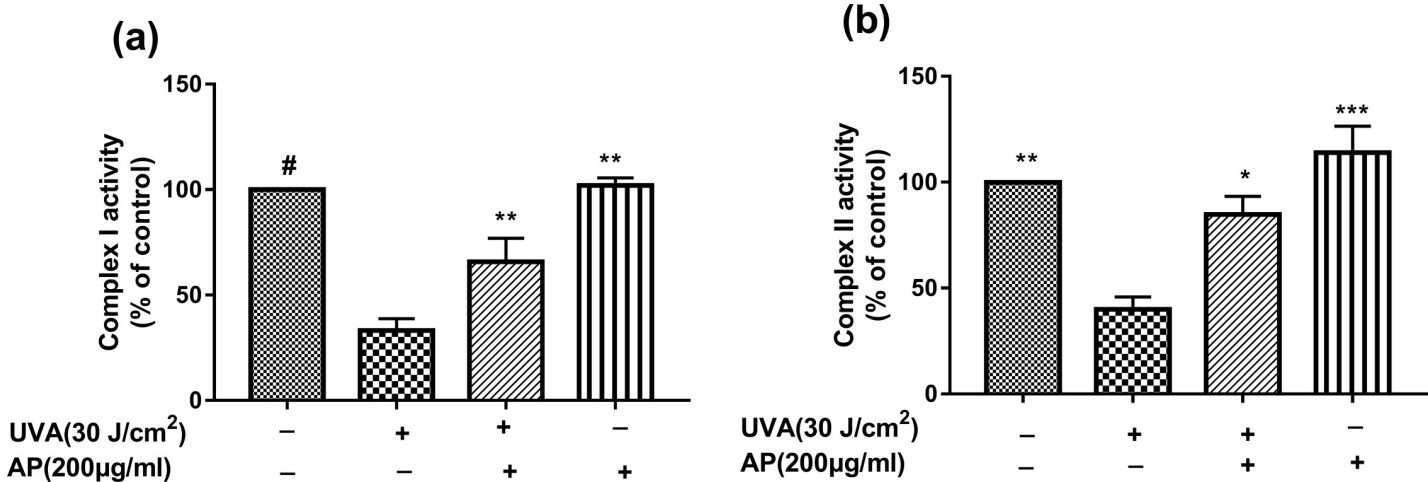

**Fig 6. AP could protect HaCaT cells against UVA-induced mitochondrial dysfunction.** HaCaT cells were seeded in 6-well plates and incubated with different concentrations of AP for 24 hours. And then, cells were treated with UVA (30J/cm$^2$). After 12 hours, the activity of mitochondrial complexes I, II were measured. (a) Determination of mitochondrial complex I activity. (b) Determination of mitochondrial complex II activity. Values are the means ± SEM of the results at least six independent experiments. $^*P$ <0.05, $^{**}P$ <0.01, $^{***}P$ < 0.001 and $^\#P$ < 0.0001 mean significant difference compared to the UVA group.

pretreatment significantly prevented the decrease in ATP levels in UVA-stimulated HaCaT cells. From Figs 3–5, we could see that 200μg/ml of AP was the most effective concentration for protecting cells from UVA-induced photo-damage in HaCaT cells.

## Effect of AP on mitochondrial functions

We next measured whether the increase in ΔΨm and ATP content by AP treatment were associated with enhanced mitochondrial function. It's showed that AP pretreatment could significantly protect HaCaT cells against UVA-induced reduction in mitochondrial complex I and complex II activities in (Fig 6). From the data showed in Figs 5 and 6, it's indicated that AP could significantly enhance mitochondrial function, which is not only related to ΔΨm, ATP levels, and complex I-II activity, but also to ROS production.

## Discussion

The skin is the largest organ in the human body. It covers the whole body and is the first barrier to protect the human body from external damage. Skin aging generally has two manifestations: natural aging and photo-damage. Natural aging mainly refers to the programmed aging caused by the irresistible factors in the body. The primary factor of extrinsic skin aging is UV radiation [31]. UVA irradiation can cause significant changes in the stratum corneum, reducing its mechanical integrity and cell cohesion. Furthermore, the UVA radiation destroys the molecular structure of cell lipids and proteins [5]. UV can effect mitochondria and generate ROS to bring about collagen degradation in human skin [38]. Some researches show that excessive ROS actuates NF-κB, which causes the level of cell surface cytokines that promotes photo-damage cytokines such as epidermal growth factor (EGF), interleukin1 (IL1), and tumor necrosis factor alpha. Moreover, these cytokines also contribute to collagen breakdown, DNA strand breaks and apoptosis of cells [39–43]. ROS can destroy DNA strands and trigger death of cells [8, 44]. As show in Figs 5(A) and 3(B), UVA irradiation resulted in a substantial increase in ROS content in HaCaT cells and decrease in cell viability, which was consistent with the former reports [38]. Furthermore, it was found that the treatment of HaCaT cells with AP significantly increased the cell viability and inhibited ROS production as shown in Figs 3

(B) and 5(A). Our results indicate that AP can remove excess ROS produced by UVA irradiation to protect HaCaT cells from UVA-induced photo-damage. Moreover, the increased ROS with age can diametrically damage the architectures of the mitochondria itself, such as proteins, lipids, and mtDNA [45–47].

UVA-induced ROS also can generate mtDNA damage, which serves as a marker of skin harm and leads to a decrease in mitochondrial energy metabolism [48]. Moreover, the senescent fibroblasts show a reduction of membrane potential [49]. As shown in Fig 4, different doses of UVA irradiation resulted in a significant decrease in MMP of HaCaT cells. However, Fig 4 shows that treatment of cells with AP significantly reverses this phenomenon. Mitochondria-induced aging diversification may affect mitochondrial respiration. Mitochondria can produce ATP, which is used as chemical energy for most eukaryotic cells. ATP produced by mitochondria also controls cell functions related to differentiation, cell signaling and cell death [50]. Subsequently, Harman established the free radical theory of aging, submitted that the mitochondria's construction of the superoxide may be middleman of FRTA [51]. This is called the mitochondrial theory of aging. In Fig 5(B), we find that UVA (30J/cm$^2$) significantly reduced ATP levels, and 200μg/ml AP pretreatment significantly prevented a decrease in ATP levels of UVA-stimulated HaCaT cells. In the present study, AP increased UVA-induced ATP expression and MMP prevented skin cells from Mitochondrial energy metabolism disorder.

Besides above, complex I, II, and III are considered to be the main sites attacked by unreasonable ROS [52–55]. Electron transfer through complexes I–IV is administered by the complexes and electron carriers such as Cytochromes C and coenzyme Q10 (CoQ10). The spread of electrons and the ATP synthesis by OxPHOS is successive within mitochondria [56]. However, the defaced mitochondria decrease in function, which in turn produces more ROS. This is known as the "vicious cycle" which is believed to produce advancing elevated levels of stress [38, 56]. The results of this study indicated that AP pretreatment could significantly protect HaCaT cells against UVA-induced reduction in mitochondrial complex I and complex II activities as shown in Fig 6. Targeting mitochondria can alleviate oxidative stress caused by UVA irradiation. Polyphenols are the most common natural products that have been observed to improve the function of stressed mitochondria, such as resveratrol which has been shown to directly impact mitochondria by modulating the oxidative phosphorylation system [57]. Moreover, the flavonoid is a polyhydroxy phenol that has photo-damage mitigating effects which are associated with preserved mitochondrial function [58, 59]. The same phenomenon has also been observed in Fig 5(A) shows that the anti-oxidative activity of AP may help in protecting the skin from UVA-induced ROS over-expression, which acts as a free radical scavenger. A previous reports clearly showed that the main reasons that influence AgNP-induced Cx43 upregulation include: changes in reactive oxygen content and activation of extracellular signal-regulated kinase and c-Jun N-terminal kinas [60]. Interestingly, we found that the AP protection mechanism is different from silver nanoparticles. In our present study, we found that AP could improve mitochondrial complex I and complex II activities to fight against the reduction of MMP and ATP, which indicates that AP could protect HaCaT cells from photo-damage via targeting to mitochondria.

## Conclusions

In conclusion, our results showed that AP treatment could effectively reduce UVA-induced ROS expression as well as improve the expression of ATP and MMP. Moreover, AP could assist HaCaT cells to fight against the reduction of ATP and MMP through elevating mitochondrial complex I and complex II activities, which indicates that AP may have the protection of skin from photo-damage by targeting to mitochondria.

## Acknowledgments

We thank LetPub (www.letpub.com) for its linguistic assistance during the preparation of this manuscript.

## Author Contributions

**Data curation:** Qiong Li.

**Project administration:** Jiejie Hao, Guangli Yu.

**Resources:** Depeng Wang, Donghui Bai, Chao Cai, Jia Li, Chengxiu Yan, Shuai Zhang, Zhijun Wu.

**Writing – original draft:** Qiong Li.

**Writing – review & editing:** Jiejie Hao, Guangli Yu.

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
