## [Decision Letter · Decision Letter 0]

22 Apr 2020

PONE-D-20-05967

Protective effects of Astragalus membranaceus polysaccharide on UVA-induced photo-aging in HaCaT cells

PLOS ONE

Dear Author,,

Thank you for submitting your manuscript to PLOS ONE. After careful consideration, we feel that it has merit but does not fully meet PLOS ONE’s publication criteria as it currently stands. Therefore, we invite you to submit a revised version of the manuscript that addresses the points raised during the review process.

We would appreciate receiving your revised manuscript by Jun 06 2020 11:59PM. To enhance the reproducibility of your results, we recommend that if applicable you deposit your laboratory protocols in protocols.io, where a protocol can be assigned its own identifier (DOI) such that it can be cited independently in the future. For instructions see: http://journals.plos.org/plosone/s/submission-guidelines#loc-laboratory-protocols

We look forward to receiving your revised manuscript.

Kind regards,

N. Rajendra Prasad, Ph.D

Academic Editor

PLOS ONE

Journal Requirements:

3. Thank you for stating the following in the Financial Disclosure section:  "This research was financially supported by National Natural Science Foundation of China and NSFC-Shandong Joint Fund for Marine Science Research Centers ( 31670811), National Science and Technology Major Project for Significant New Drugs Development (2018ZX09735004), Fundamental Research Funds for the Central Universities (201762002).The funders had no role in study design, data collection and analysis, decision to publish, or preparation of the manuscript."

We note that one or more of the authors are employed by a commercial company: 'Infinitus (China) Company Ltd., Guangzhou 510600, China'

Additional Editor Comments :

• Give the structure of the isolated compound.

• The reported photoprotection is due to scavenging UVA-induced ROS or the sunscreen property of the compound?. The authors may have to report the SPF value of the isolated compound.

• How do the authors irradiated the HaCaT cells? How many cells used for irradiation. Give the brief methodology of the UVA treatment and/or the polysaccharide treatment.

• The authors have not analyzed UVA-mediated photoaging specific markers like matrix metalloproteinases expression or collagen degradation in this study. Therefore, the title may not be like “UVA-induced photoaging”. Either include effect of AP on certain photoaging markers or re-write the title as “Photoprotective effect of AP….”

• Additional experiments on the effect of AP against UVA-induced mitDNA damage may further improve the novelty of the present work.

Reviewers' comments:

Reviewer's Responses to Questions

**Comments to the Author**

1. Is the manuscript technically sound, and do the data support the conclusions?

Reviewer #1: Yes

2. Has the statistical analysis been performed appropriately and rigorously? 

Reviewer #1: Yes

3. Have the authors made all data underlying the findings in their manuscript fully available?

Reviewer #1: Yes

4. Is the manuscript presented in an intelligible fashion and written in standard English?

Reviewer #1: Yes

5. Review Comments to the Author

Reviewer #1: This is an interesting article on the effect of Astragalus membranaceus polysaccharide (AP) on photo-aging of HaCaT cells by reducing UVA-induced intracellular reactive oxygen species (ROS) production and mitochondrial membrane potential, thereby altering ATP content. The authors could also refer to the literature on the positive effect of silver nanoparticles on the reduction of UV-induced damage to HaCaT cells and comment on whether the AP protective mechanism is different from that of silver nanoparticles.

6. PLOS authors have the option to publish the peer review history of their article (what does this mean?). If published, this will include your full peer review and any attached files.

Reviewer #1: No

---

## [Author Response · Author response to Decision Letter 0]

11 May 2020

Dear Editors:

On behalf of my co-authors, we thank you very much for giving us an opportunity to revise our manuscript. And the positive and constructive comments from editor and reviewers on our manuscript entitled “Protective effects of Astragalus membranaceus polysaccharide on UVA-induced photo-aging in HaCaT cells”(ID: PONE-D-20-05967) were highly appreciated These comments are all valuable and very helpful for revising and improving our paper, as well as the important guiding significance to our researches. We have studied comments carefully and have made correction which we hope to meet with approval. Revised portion are marked in red with underline in the paper. We have updated Funding Statement and Competing Interests Statementaccording to all PLOS ONE policies. The main corrections in the paper and the responds to the reviewer’s comments are listed as follows. We would like to express our great appreciation to you and reviewers for comments on our paper again. And we are Looking forward to hearing from you.

Yours sincerely,

Jiejie Hao

E-mail: 2009haojie@ouc.edu.cn

Additional Editor Comments :

• Give the structure of the isolated compound.

• The reported photoprotection is due to scavenging UVA-induced ROS or the sunscreen property of the compound?. The authors may have to report the SPF value of the isolated compound.

• How do the authors irradiated the HaCaT cells? How many cells used for irradiation. Give the brief methodology of the UVA treatment and/or the polysaccharide treatment.

• The authors have not analyzed UVA-mediated photoaging specific markers like matrix metalloproteinases expression or collagen degradation in this study. Therefore, the title may not be like “UVA-induced photoaging”. Either include effect of AP on certain photoaging markers or re-write the title as “Photoprotective effect of AP….”

• Additional experiments on the effect of AP against UVA-induced mitDNA damage may further improve the novelty of the present work.

Response: Special thanks for your comments. Your suggestions are very valuable and helpful to our further study. 

1.We are greatly appreciated your professional comments. we provide the structure of the isolated compound.

 Additional Editor Comments :

• Give the structure of the isolated compound.

• The reported photoprotection is due to scavenging UVA-induced ROS or the sunscreen property of the compound?. The authors may have to report the SPF value of the isolated compound.

• How do the authors irradiated the HaCaT cells? How many cells used for irradiation. Give the brief methodology of the UVA treatment and/or the polysaccharide treatment.

• The authors have not analyzed UVA-mediated photoaging specific markers like matrix metalloproteinases expression or collagen degradation in this study. Therefore, the title may not be like “UVA-induced photoaging”. Either include effect of AP on certain photoaging markers or re-write the title as “Photoprotective effect of AP….”

• Additional experiments on the effect of AP against UVA-induced mitDNA damage may further improve the novelty of the present work.

Response: Special thanks for your comments. Your suggestions are very valuable and helpful to our further study. 

1.We are greatly appreciated your professional comments. we provide the structure of the isolated compound.

 Fig S1. The structure of AP

2.Great question! It’s valuable to investigate the intrinsic photoprotection mechanisms of AP we observed in our experiments. So we have planned to compare the protective effects between the present AP and enzyme-degraded AP oligosaccharides, and designed to probe both of them by FITC and observe their process of entering the skin cells. Meanwhile, we also designed to detect the ROS production level along with the time of extra- and inter-cells. We hope these experiments would help us know the question you bring for us. Thanks again. 

3. First, we planted 8000 cells per well in a 96-well plate and placed it in a constant temperature cell incubator for 12 hours; then, added different concentrations of AP and placed it in a constant temperature cell incubator for 48 hours. After the incubation, the 96-well plate was placed in a UV cross-linker and irradiated with 30J/cm2 UVA; after the irradiation was completed, the medium was replaced and placed in a constant temperature cell incubator for 12 hours, and then, the biochemical indicators were measured.

4. Thanks a lot for your helpful and constructive suggestions. We have re-write the title as “Photoprotective effect of Astragalus membranaceus polysaccharide on UVA-induced damage in HaCaT cells”

5. We are greatly appreciated your professional comments. We will detect the UVA-induced mtDNA damage in our further investigation to compare the difference between the polysaccharides of AP and its oligosaccharides. 

Reviewer #1: This is an interesting article on the effect of Astragalus membranaceus polysaccharide (AP) on photo-aging of HaCaT cells by reducing UVA-induced intracellular reactive oxygen species (ROS) production and mitochondrial membrane potential, thereby altering ATP content. The authors could also refer to the literature on the positive effect of silver nanoparticles on the reduction of UV-induced damage to HaCaT cells and comment on whether the AP protective mechanism is different from that of silver nanoparticles.

Response: Thanks very much for your suggestion. We have made additions marked in red in the revised manuscript according to your suggestion.

Response: Special thanks for your comments. We have updated Funding Statement and Competing Interests Statementaccording to all PLOS ONE policies in line 338-344 and line 349-352.

Funding

This work was supported by National Natural Science Foundation of China (81973231, 81402982, 31670811), National Science and Technology Major Project for Significant New Drugs Development (2018ZX09735004). The funders had no role in study design, data collection and analysis, decision to publish, or preparation of the manuscript.

Competing interests

The authors of Qiong Li, Depeng Wang, Donghui Bai , Chao Cai, Jia Li, Chengxiu Yan, Shuai Zhang, Jiejie Hao and Guangli Yu declare that they have no competing interests; Zhiyun Wu also provides a Competing Interests Statement declaring that she has no competing interests.

---

## [Editor Report · Decision Letter 1]

17 Jun 2020

Photoprotective effect of Astragalus membranaceus polysaccharide on UVA-induced damage in HaCaT cells

PONE-D-20-05967R1

Dear Dr. Hao,

We’re pleased to inform you that your manuscript has been judged scientifically suitable for publication and will be formally accepted for publication once it meets all outstanding technical requirements.

Kind regards,

N. Rajendra Prasad, Ph.D

Academic Editor

PLOS ONE
---

## [Editor Report · Acceptance letter]

8 Jul 2020

PONE-D-20-05967R1 

Photoprotective effect of Astragalus membranaceus polysaccharide on UVA-induced damage in HaCaT cells 

Dear Dr. Hao:

I'm pleased to inform you that your manuscript has been deemed suitable for publication in PLOS ONE. Congratulations! Your manuscript is now with our production department. 

Kind regards, 

on behalf of

Dr. N. Rajendra Prasad 

Academic Editor

PLOS ONE